# Population-level genome-wide STR discovery and validation for population structure and genetic diversity assessment of *Plasmodium* species

Jiru Han[1,2], Jacob E. Munro[1,2], Anthony Kocoski[1,3], Alyssa E. Barry[1,2,4,5], Melanie Bahlo[1,2]*

1 Population Health and Immunity Division, The Walter and Eliza Hall Institute of Medical Research, Melbourne, Australia, 2 Department of Medical Biology, The University of Melbourne, Melbourne, Australia, 3 Department of Mathematics and Statistics, The University of Melbourne, Melbourne, Australia, 4 Disease Elimination Program, Burnet Institute, Melbourne, Australia, 5 IMPACT Institute for Innovation in Mental and Physical Health and Clinical Translation, Deakin University, Geelong, Australia

* bahlo@wehi.edu.au

**Data Availability Statement:** Data used in this study are available from the MalariaGEN Plasmodium falciparum Community Project

## Abstract

Short tandem repeats (STRs) are highly informative genetic markers that have been used extensively in population genetics analysis. They are an important source of genetic diversity and can also have functional impact. Despite the availability of bioinformatic methods that permit large-scale genome-wide genotyping of STRs from whole genome sequencing data, they have not previously been applied to sequencing data from large collections of malaria parasite field samples. Here, we have genotyped STRs using HipSTR in more than 3,000 *Plasmodium falciparum* and 174 *Plasmodium vivax* published whole-genome sequence data from samples collected across the globe. High levels of noise and variability in the resultant callset necessitated the development of a novel method for quality control of STR genotype calls. A set of high-quality STR loci (6,768 from *P. falciparum* and 3,496 from *P. vivax*) were used to study *Plasmodium* genetic diversity, population structures and genomic signatures of selection and these were compared to genome-wide single nucleotide polymorphism (SNP) genotyping data. In addition, the genome-wide information about genetic variation and other characteristics of STRs in *P. falciparum* and *P. vivax* have been available in an interactive web-based R Shiny application PlasmoSTR (https://github.com/bahlolab/PlasmoSTR).

## Author summary

Malaria is a severe disease caused by a genus of parasites called *Plasmodium* and is transmitted to humans through infected *Anopheles* mosquitoes. *P. falciparum* and *P. vivax* are the predominant species responsible for more than 95% of all human malaria infections which continue to pose a significant challenge to human health. Antimalarial drug resistance is a serious threat hindering the elimination of malaria. As such, it is important to

(https://www.malariagen.net/resource/26), MalariaGEN Pf3k genetic cross dataset (https://www.malariagen.net/data/pf3K-5), and Plasmodium vivax Genome Variation project (https://www.malariagen.net/projects/p-vivax-genome-variation) and P. vivax data from the Broad Institute, as described in Hupalo et al., Nature Genetics 2016 (https://doi.org/10.1038/ng.3588). The R code of the multivariable logistic regression modeling and genome-wide information of STRs in P. falciparum and P. vivax is available on GitHub (https://github.com/bahlolab/PlasmoSTR). All other relevant data are within the manuscript and Supporting information.

**Funding:** This work was supported by an NHMRC Senior Research Fellowship to M.B [Grant number: 1102971] (https://www.nhmrc.gov.au). J.H was supported by a Melbourne Research Scholarship (The University of Melbourne, https://www.unimelb.edu.au) and a WEHI PhD Scholarship (The Walter and Eliza Hall Institute of Medical Research, https://www.wehi.edu.au). This work was also made possible through the Victorian State Government Operational Infrastructure Support and Australian Government National Health and Medical Research Council (NHMRC) Independent Research Institute Infrastructure Support Scheme (IRIISS). The funders had no role in study design, data collection and analysis, decision to publish, or preparation of the manuscript.

**Competing interests:** The authors have declared that no competing interests exist.

understand the role of genomic variation in the development of antimalarial drug resistance. STRs are an important source of genomic variation that, from a population genetics perspective, have several advantages over SNPs, including being highly polymorphic, having a higher mutation rate, and having been widely used to study population structure and genetic diversity. However, STRs are not routinely genotyped with bioinformatic tools across the whole genome with short read sequencing data because they are difficult to identify and genotype accurately, as they vary in size and may align poorly to the reference genome, therefore requiring rigorous quality control (QC). In this study, we genotype STRs using HipSTR in more than 3,000 *P. falciparum* and 174 *P. vivax* whole-genome sequence samples collected world-wide. We develop a multivariable logistic regression model for the measurement and prediction of the quality of STRs. In addition, we use a set of genome-wide high-quality STRs to study parasite population genetics and compare them to genome-wide SNP genotyping data, revealing both high consistency with SNP based signals, as well as identifying some signals unique to the STR marker data. These results demonstrate that the identification of highly informative STR markers from large numbers of population samples is a powerful approach to study the genetic diversity, population structures and genomic signatures of selection in *P. falciparum* and *P. vivax*. Furthermore, we built an interactive web-based R Shiny application PlasmoSTR (https://github.com/bahlolab/PlasmoSTR) that includes genome-wide information about genetic variation and other characteristics of the high quality STRs identified in *P. falciparum* and *P. vivax*, allowing researchers to explore and visualize the specific STRs.

## Introduction

Short tandem repeats (STRs), also known as microsatellites, are tandem nucleotide repeats (1–9 base pairs, bp) that are both abundant throughout the genome and highly polymorphic. Unlike many other types of genetic markers, STRs have a high mutation rate that is highly variable across different loci. *P. falciparum* has the most AT-rich eukaryotic genome known, with 80.6% A + T content overall and approaching 90% in introns and intergenic regions [1]. As a consequence, many regions in *P. falciparum* genome are highly repetitive, and STRs are found in abundance in both coding and noncoding regions throughout the *P. falciparum* parasite genome [1,2] leading to about 10.74% of the *P. falciparum* genome being composed of STRs [1,3]. In contrast, the total A+T content in *P. vivax* is 57.7% [4,5]. In organisms with AT content close to 50%, such as *Drosophila* or humans, STRs only account for 1–3% of the genome [3,6,7]. These repetitive sequences can arise, expand or contract rapidly. In many cases, the simple homopolymer repeats tend to evolve neutrally and may not have a function, representing non-functional 'junk DNA', however more complex sequences seem to be under selective pressure indicating a functional role [2,8,9]. The repetitive protein sequences of *Plasmodium* have been previously shown to alter protein activity, protein folding efficiency, stability, or aggregation and play an important role in the formation of key structural elements of protein function [9]. STRs in coding regions with a motif size that is a multiple of three (e.g. trinucleotide or hexanucleotide repeats) will not result in a frame-shift mutation when repeats are deleted or added, but can change protein sequences [10]. For example, the *Pfnhe-1* protein contains a polymorphic amino acid motif DNNND (GATAACAATAATGAT) and DDNHNDNHNND (GATGATAACCATAATGATAATCATAATAATGAT) which affects the *P. falciparum* Na+/H+ exchanger capabilities, and influences quinine resistance by combining *Pfcrt* and *Pfmdr1* [11,12].

STRs have also been widely used to study the population structure and genetic diversity of *P. falciparum* and *P. vivax* populations in many countries [13–16]. However, most studies used relatively few (< 20) polymorphic STR markers. These STRs are generally assumed to have putative neutrality. Making inferences about genetic structure from a limited number of STRs can be deemed to be insufficient [17]. These STRs were typed using a variety of low-throughput lab-based methods, most recently with capillary electrophoresis [18,19]. Extending these low-throughput methods genome-wide or even to a few hundred STR markers, is prohibitive in both time and cost. A recent study used GATK's HaplotypeCaller [20] to identify 3,168,721 SNPs and 2,882,975 indels (or SNP/indel combinations) in 7,182 samples (part of the data used in the current study) [21]. However, the population genetic analyses in this study only assessed the genetic variation from the SNPs but ignored the indels that might overlap with some STRs. Previous studies of genome-wide STRs analyses either examined just the *Plasmodium* reference genome or a limited number (less than 300) of *in vitro Plasmodium* samples and mainly focused on the composition and function of STRs [9,10,22] and their mutation rates [3,23]. The overall contributory effect of STR variation in *Plasmodium* field samples has not been evaluated at a genome-wide level. Therefore, characterization of STRs in the larger *Plasmodium* cohorts, from field samples, would be useful for the understanding of genetic diversity, and environmental and evolutionary adaptations of STRs in *Plasmodium*.

Recently developed bioinformatic methods that infer the length of STR alleles using short-read sequencing data permit STR genotyping from large collections of samples. There are many tools for genotyping STRs, such as LobSTR [24], RepeatSeq [25], HipSTR [26] and GangSTR [27]. The widely used variant calling tool GATK HaplotypeCaller [20] can also be used for finding and calling a limited set of STRs, but it is specifically designed for SNPs/indel calling with the indel calling have lower accuracy compared to SNP calling [28]. We used HipSTR [26], which is a haplotype-based method specifically designed for STR analysis. While other STR tools were mainly developed for calling the STR length per individual sample, HipSTR considers the entire sequence across all samples in the dataset for each STR site, and has been shown to outperform other tools when considering genotyping error rate, even with low coverage [28,29].

Here, we report the largest to date STR typing study in more than 3,000 *P. falciparum* [21] and 174 *P. vivax* [30,31] short-read whole genome sequencing samples sourced from global malaria hot-spots. A central aim of this work was to develop a filtering strategy to discover a set of high-quality STR variants and build a publicly available and easy to use resource available for researchers who are interested in looking at the role of specific STRs throughout the *Plasmodium* genome. We then aimed to compare the performance of genome-wide SNPs data and STRs data in the following aspects: delineate population structure, genetic diversity, and genetic differentiation metrics. We also explored the biological importance of STR variation in different populations and identified STR loci that may be linked to antimalarial drug resistance.

## Results

### SNP genotyping

**MalariaGEN global *P. falciparum* dataset.** Variations at more than three million positions were discovered in the *P. falciparum* dataset in the first stage of variant analysis. These included 1,542,905 SNPs and 1,545,263 indels. After performing all the filtering procedures (see Materials and Methods for more details), a total of 213,757 biallelic SNPs were retained. We removed samples with multiple infections as determined by the within-host infection fixation index ($F_{ws}$) metric [32,33], or higher than 10% missing genotypes or other quality control

issues, leaving a total of 3,047 high-quality samples from the 26 countries (excluding Burkina Faso) from 8 populations. Sample size varied by population with South America (SAM) = 31, West Africa (WAF) = 959, Central Africa (CAF) = 100, East Africa (EAF) = 327, South Asia (SAS) = 32, the western part of Southeast Asia (WSEA) = 690, the eastern part of Southeast Asia (ESEA) = 827 and Oceania (OCE) = 81 [21]. The downstream analysis of the *P. falciparum* dataset in this study is based on the filtered dataset of high-quality SNP genotypes in 3,047 samples.

**Global *P. vivax* dataset.**   Variations included 1,345,364 SNPs and 715,369 indels discovered in the *P. vivax* dataset in the first stage of variant analysis. After performing all the filtering procedures, a total of 188,571 biallelic SNPs were retained. We removed samples with multiple infections, or higher than 10% missing genotypes or other quality control issues, leaving 174 high-quality samples from 11 countries. Sample size varied by country with Brazil = 2, Cambodia = 16, Colombia = 27, Indonesia = 2, Malaysia = 3, Mexico = 17, Myanmar = 7, Peru = 30, Papua New Guinea (PNG) = 7, Thailand = 59, Vietnam = 4 samples respectively. The downstream analysis of the *P. vivax* dataset in this study is based on the filtered dataset of high-quality SNP genotypes in 174 samples.

## STR genotyping

We identified 104,649 high quality STRs from the *P. falciparum* Pf3D7 reference genome (accounting for 9.29% of the genome) and 40,224 STRs from the *P. vivax* PvP01 reference genome (accounting for 3.16% of the genome) by using Tandem Repeats Finder (TRF) [34]. The number of STRs in *P. falciparum* is almost three times that of *P. vivax*. STRs with a 1–6 bp repeat unit account for 97.32% of the *P. falciparum* Pf3D7 reference genome and 95.27% of the *P. vivax* PvP01 reference genome STRs. Of these, homopolymeric tracts account for 40.09% of Pf3D7, and 64.16% of PvP01. The dinucleotide repeats account for 24.66% of Pf3D7, while only 6.38% of PvP01. The higher proportion of dinucleotide repeats in *P. falciparum* can be attributed to the overall high AT content of the *P. falciparum* genome, with 24% of dinucleotide repeats in *P. falciparum* having the 'AT' motif, and 3% of dinucleotide repeats in *P. vivax* having the 'AT' motif.

A total of 20,196 STRs remained for downstream analysis across 3,047 *P. falciparum* samples after the initial QC filtering steps (see Materials and Methods for more details). The majority of STRs were located in the promoter region (53.02%), coding region (25%), followed by the intergenic region (12.91%), intron region (8.61%) and other regions. *P. falciparum* is an extremely AT-rich genome but with higher GC content in coding and promoter regions, probably leading to more confident calling and higher quality STRs in those regions. The mononucleotide repeats appeared to be the most abundant STRs in *P. falciparum* (9,382, 46.45%), followed by trinucleotide repeats (3,767, 18.65%) and dinucleotide repeats (3,563, 17.64%).

A total of 23,146 STRs were retained for downstream analysis across 174 *P. vivax* samples after performing all the filtering procedures. The number of STRs varies in different genomic regions: promoter region (48.88%), coding region (27.35%), intergenic region (16.12%), followed by the intron region (7.47%) and other regions. The mononucleotide repeats appeared to be the most abundant STRs in *P. vivax* (14,386, 62.15%), followed by trinucleotide repeats (4,251, 18.37%) and dinucleotide repeats (1,324, 5.72%).

To evaluate the accuracy of HipSTR STR genotyping, we assessed the Mendelian error rate (MER) of STRs in three previously published *P. falciparum* genetic crosses [35–37]. We obtained an average MER of 4.53% in the 3D7×HB3, 2.08% in the HB3×Dd2, and 2.99% in the 7G8×GB4 genetic crosses respectively. In addition, the average MER in the core genome region is lower than in the non-core genome region in all three genetic crosses (S1 Fig). As

expected, the average MER of STR genotyping is higher than the MER of the SNP genotyping (S1 Fig). We then evaluated the genotype discordance rate of STRs and obtained an average discordance rate of 4.34% in the 3D7×HB3 and 5.60% in the 7G8×GB4 genetic crosses respectively, approximately 3-fold higher than the SNP allele discordance rate (S2 Fig). We then checked the genotype accuracy of the high-quality and low-quality STRs defined by our model (see Materials and Methods for more details) in the same dataset (Pf3k genetic cross). We obtained an average MER of 0.45% in the 3D7×HB3, 0.4% in the HB3×Dd2, and 0.65% in the 7G8×GB4 genetic crosses for the high-quality STRs, while the low-quality STRs had an error rate at least two-fold higher (S3A Fig). A similar pattern was also observed for the genotype discordance rates between replicates (S3B Fig). In addition, the MER and genotype discordance between replicates of high-quality and low-quality STRs both were lower than for the overall genetic crosses STRs (S3 Fig). The Pf3k genetic cross dataset has a higher sequencing mean coverage (~71.8×) compared with the 3,047 field samples (~54.1×) from the Malaria-GEN *Plasmodium falciparum* Community Project [21]. This higher sequencing coverage will increase the number of spanning reads [26] and leads to a set of genetic crosses STRs that were not successfully genotyped in the 3,047 field samples and also resulted in a higher MER and genotype discordance rates for the genetic crosses data. The number of segregating sites between pairwise samples of STRs and SNPs demonstrated a highly significant positive correlation (Pearson's r = 0.9785; $P < 2.2 \times 10^{-16}$) (S4A Fig). The relative number of STR segregating sites (segregating sites/No. of variants) between pairwise samples was higher than SNPs, indicating the high mutability of the STRs (S4B Fig). To measure HipSTR's quality of prediction, we also inspected HipSTR's genotype calls for the markers against the gel electrophoresis (GE) calls (see Materials and Methods for more details) and found HipSTR is calling length polymorphisms accurately with respect to the GE calls (S5 Fig and S1 Table).

**Multivariable logistic regression modeling for measurement and prediction of the quality of STRs.** A set of metrics including QC metrics from HipSTR and other metrics that were deemed useful were derived and used for the prediction of the STR quality. These are summarised in S2 Table. A multivariable logistic regression analysis was then performed to identify potential predictors of STR quality (see Materials and Methods for more details). Because STRs with mononucleotide repeats are more abundant and have higher error rates, we built separate regression models for the mononucleotide (1 bp motif) STRs and the polynucleotide (2–9 bp motif) STRs. Examination of Spearman's correlation coefficients ($R^2$) suggested that selecting the first five SNP principal components (PCs) were sufficient to capture the signal STRs. For the *P. falciparum* dataset, features that were significantly associated with the STR quality and the results obtained for the estimated coefficients for both the mononucleotide STR and polynucleotide STR models are presented in Table 1. For both the mononucleotide STR and polynucleotide STR models, the STR quality tends to be more associated with the STR features that capture population specific aspects of the dataset. Compared to the polynucleotide STR model, 'Mean_Posterior' which is the mean posterior probability of the STR genotype across all samples derived from HipSTR, exhibited a large effect, but only in the mononucleotide STR model. The closer this quantity is to 1, the higher the confidence of the called genotype. We also built the model for the *P. falciparum* dataset that did not use the reported population origins of each sample as some samples were clearly distinct to the majority of samples from a particular country. The model of the *P. falciparum* sample without the population origin label was found to produce very similar results (S3 Table). For all further analysis only the model with the given population label was used.

We also compared the prediction performance of the *P. falciparum* complete dataset with five-fold cross-validation (see Materials and Methods). The performance of each dataset is compared in terms of their receiver–operator–characteristic (ROC) curves and the area-

**Table 1. Multivariable logistic regression model's estimated coefficients and respective 95% confidence intervals.**

|  | Variable | Coefficient estimate | Lower 95% CI | Upper 95% CI | *P* values |
|---|---|---|---|---|---|
| **Modeling Mononucleotide STR** | (Intercept) | 0.48 | 0.38 | 0.59 | <0.001 |
|  | Repeat units | 0.12 | 0.05 | 0.19 | <0.001 |
|  | GC_Diff | 0.09 | 0.02 | 0.16 | 0.01 |
|  | Missingness | 0.06 | -0.01 | 0.12 | 0.1 |
|  | Mean_Posterior | 0.81 | 0.70 | 0.92 | <0.001 |
|  | Mean_stutter | -0.15 | -0.24 | -0.07 | <0.001 |
|  | He | 7.00 | 6.38 | 7.64 | <0.001 |
|  | MeanHe | -3.87 | -4.57 | -3.17 | <0.001 |
|  | MinimumHe | -0.81 | -1.03 | -0.57 | <0.001 |
|  | MaximumHe | 2.33 | 2.08 | 2.59 | <0.001 |
| **Modeling Polynucleotide STR** | (Intercept) | 1.55 | 1.40 | 1.71 | <0.001 |
|  | Length | -0.11 | -0.18 | -0.04 | 0.003 |
|  | Repeat units | 0.16 | 0.06 | 0.26 | 0.001 |
|  | GC_Flank | 0.11 | 0.05 | 0.17 | <0.001 |
|  | Mean_Posterior | 0.10 | 0.00 | 0.20 | 0.05 |
|  | Mean_Stutter | -0.11 | -0.20 | 0.03 | 0.04 |
|  | He | 10.20 | 9.02 | 11.41 | <0.001 |
|  | MeanHe | -12.41 | -13.73 | -11.10 | <0.001 |
|  | JostD | -1.10 | -1.42 | -0.75 | <0.001 |
|  | MinimumHe | -0.48 | -0.84 | -0.12 | 0.009 |
|  | MaximumHe | 7.02 | 6.52 | 7.52 | <0.001 |

The model was fitted on the *P. falciparum* dataset.

under-the-curve (AUC). The AUC values of the *P. falciparum* complete dataset is 0.9132 in the mononucleotide STR model and 0.9527 in the polynucleotide STR model, and these five validation-datasets range from 0.7911 to 0.8209 in the mononucleotide STR model, and 0.8853 to 0.8998 in the polynucleotide STR model (Fig 1). The performance fluctuation depends on the size of the datasets. The predictive performance in the whole dataset was generally superior to that in the smaller training datasets (S6 and S7 Figs). This was observed in both the *P. falciparum* mononucleotide STR and polynucleotide STR models, suggesting that larger datasets may

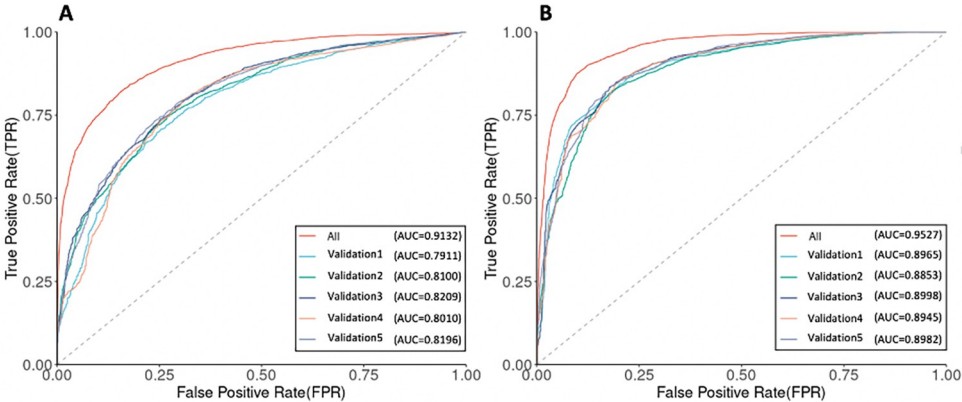

**Fig 1. ROC curves and AUC values of the *P. falciparum* complete dataset and five validation-datasets.** (A) The mononucleotide STR model. (B) The polynucleotide STR model.

improve prediction power. In this work, we select the whole *P. falciparum* dataset with a higher AUC value to perform all subsequent analyses. The model of the *P. falciparum* sample without the population origin label also showed high performance for both the mononucleotide STR (AUC = 0.9198) and polynucleotide STR (AUC = 0.9589). The *P. falciparum* model was observed to be very stable, with regards to which measures of quality were used in the model, and reproducible, with the five-fold cross validation sets giving very similar results in both the mononucleotide and polynucleotide STR models.

For the *P. vivax* dataset, features that were most significantly associated with the STR quality are summarised in S4 Table. The *P. vivax* model also showed high performance for both the mononucleotide STR (AUC = 0.9186) and polynucleotide STR (AUC = 0.9548). For the *P. vivax* mononucleotide STR and polynucleotide STR models, the STR quality is also more associated with the STR features that capture population specific aspects of the dataset, showing similar results as *P. falciparum*. However one of the STR variables, MinimumHe, displayed an opposite relationship compared to the *P. falciparum* STR models. For *P. vivax*, four countries have few samples ($< 5$), which may affect the robustness of the MinimumHe estimate and thus may have led to this difference. To investigate this, we also built the model of the *P. vivax* sample without the population origin labels, wherein the smallest group was now 23 samples. The results obtained for the estimated coefficients for both the mononucleotide STR and polynucleotide STR models are in S5 Table. The model of the *P. vivax* sample without the population origin label also showed high performance for both the mononucleotide STR (AUC = 0.9295) and polynucleotide STR (AUC = 0.9596). It was found to produce very similar results for the statistically significantly ($P < 0.001$) associated STR variables, and additionally, the MinimumHe variable showed the same negative relationship as in the *P. falciparum* STR models.

Based on the predicted probability of the logistic regression model, for the *P. falciparum* mononucleotide STR model, we select the predicted probability greater than 0.6 as the cut-off value to retain high-quality STRs, while for the *P. falciparum* polynucleotide STR model we chose 0.8 (see Materials and Methods for more details). A total of 6,768 high-quality STR (2,563 mononucleotide STR and 4,205 polynucleotide STR) loci were thus selected. The high-quality STRs can be accessed at https://github.com/bahlolab/PlasmoSTR through a web-based application for interactive data exploration and visualization. The STRs with a 1–3 bp repeat unit in the 3,047 *P. falciparum* 3D7 samples account for 83.05% of all retained STRs. Of these, homopolymeric tracts account for 37.87%. The dinucleotide repeats account for a further 26.06% (Fig 2A). The motif size of STRs showed differential distribution among various genomic features (Fig 2B). The frequency of the trinucleotide repeats is higher in coding regions than in intronic, intergenic, and promoter regions, which has been previously observed in other species, including humans and is an example of survivorship bias with non 3-mer motifs likely to disrupt the transcript and be deleterious causing strong selection against such STRs in coding regions [38–40]. Similar distributions were also found in introns and intergenic regions, except for the mononucleotide STRs which are enriched in intergenic regions. Promoters also show a difference in the proportions of different motif sizes compared to the non-coding regions with an abundance of highly polymorphic dinucleotide (2 bp) STRs.

For the *P. vivax* mononucleotide STR model, we selected the predicted probability greater than 0.6 as the cut-off value to retain high-quality STRs, while for the *P. vivax* polynucleotide STR model we chose 0.2. A total of 3,496 high-quality STR (1,648 mononucleotide STR and 1,848 polynucleotide STR) loci were therefore selected (Fig 2C and 2D). Compared with *P. falciparum*, *P. vivax* has fewer 2 bp repeats and more 1, 8 and 9 bp repeats. The distribution of high quality STRs motif sizes (1–9 bp) of the 3,047 *P. falciparum* samples and 174 *P. vivax* samples were significantly associated with the motif size composition of the reference genome (Chi-square test, *P. falciparum*: $P < 2.2 \times 10^{-16}$; *P. vivax*: $P < 2.2 \times 10^{-16}$), as well as the

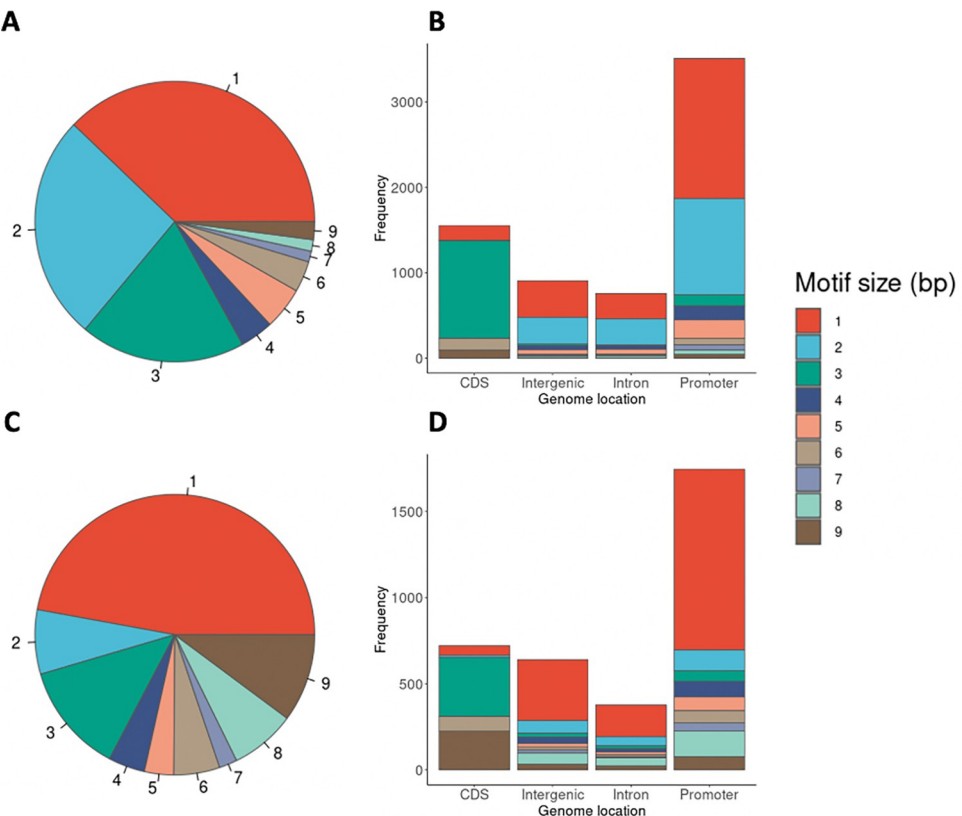

**Fig 2. STR mutation distributions of the *P. falciparum* and *P. vivax* samples.** (A) Distribution of motif sizes (1–9 bp) of the 3,047 *P. falciparum* samples, colored by the motif size. (B) Distribution of motif size among various genomic features of the 3,047 *P. falciparum* samples. CDS represents coding DNA sequences. (C) Distribution of motif sizes of the 174 *P. vivax* samples. (D) Distribution of motif size among various genomic features of the 174 *P. vivax* samples. The genomic features are labeled along the X-axis for (B) and (D). The frequencies of each motif size are calculated as the total bases covered by STRs of a given motif size divided by the total bases covered by all STRs, labeled along the Y-axis.

distribution of STRs among various genomic features (Chi-square test, *P. falciparum*: $P = 0.0059$; *P. vivax*: $P < 1.32 \times 10^{-6}$).

**Population structure analysis.** We investigated the population genetic structure of the global *P. falciparum* and *P. vivax* parasite population by performing dimensionality reduction analyses, applying both uniform manifold approximation and projection (UMAP) and principal component analysis (PCA), and generating neighbor joining trees (NJTs) for all *P. falciparum* and *P. vivax* samples based on the SNP and STR genotypes. For the *P. falciparum* dataset, UMAP on the top five PCs of both SNP and STR genotypes can distinguish the SAM, OCE, SAS, Asia (WSEA, ESEA), and Africa (WAF, CAF, EAF) parasite populations from different geographic regions, with each of these populations being more strongly differentiated from all other populations, but the SNP data from the WSEA and ESEA populations suggest greater genetic similarity between these populations than the matching STR data. Conversely, the STR data of African sub-regions appear to be genetically more similar than the corresponding SNP data suggests (Fig 3A and 3B). All population subdivisions supported by the UMAP analyses were also present in the PCA analysis (S8 Fig). In general, the global *P. falciparum* parasite population formed four distinct clusters: SAM, Africa (WAF, CAF, EAF), OCE, and the Asia (WSEA, ESEA) region. This clustering may be affected by a variety of factors, including vector

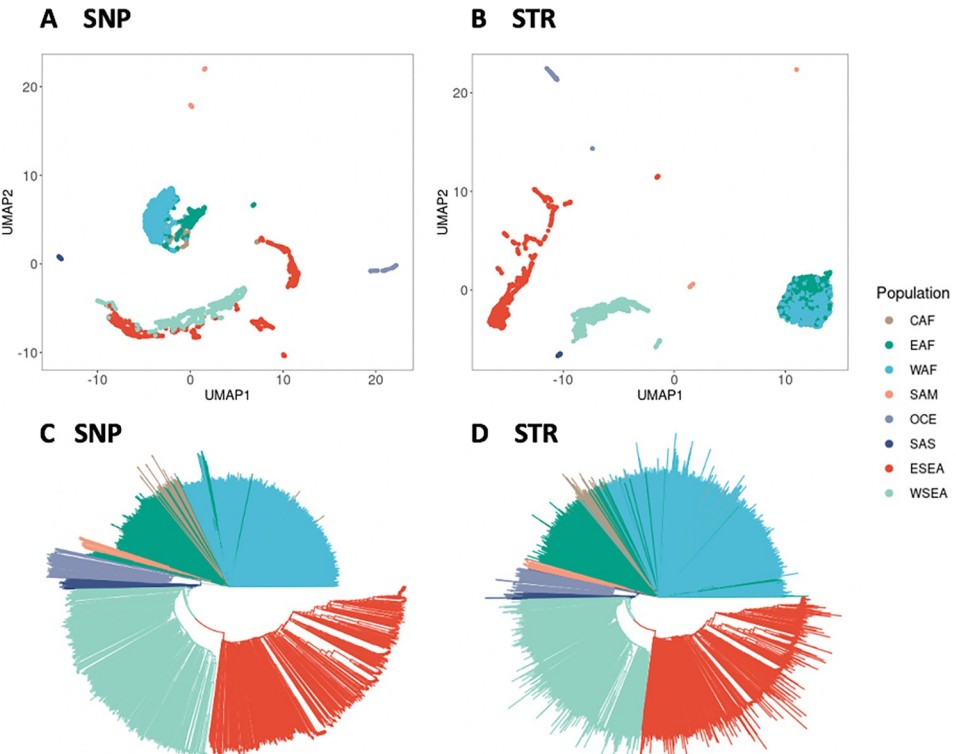

**Fig 3. Population structure analysis of the 3,047 *P. falciparum* samples of SNP and STR data.** (A) UMAP clustering of the top five principal components of the SNP data, colours representing the eight different populations. (B) UMAP clustering of the top five principal components of the STR data. (C) Neighbor joining tree based on the SNP data. (D) Neighbor joining tree based on the STR data. Branches are colored according to the population.

species, varying malaria transmission intensity, and the historical usage of antimalarial drugs, all of which are confounded by the time of collection of the samples. To further explore clustering patterns and investigate the average genetic dissimilarity between pairs of individuals, phylogenetic analysis was performed to produce a neighbor joining tree. The neighbor joining trees also recapitulate the population structure from the clustering analyses (Fig 3C and 3D).

For the sub-population structure analysis from the *P. falciparum* dataset, we found that Ethiopia was genetically distinct from other EAF countries, and that the two countries of Colombia and Peru in the SAM population were also genetically distinct (S9 Fig). This was observed in both the STR and SNP data. We then explored the local levels of parasite population structure and found that the STR data showed that the Milne Bay site in PNG formed a well-separated and distinct cluster with two other remote PNG sites, East Sepik and Madang, while the SNP data was unable to separate these clusters (S10 Fig). Additionally, the STR data showed a well-separated cluster in two different sample collection sites in Laos (Attapeu and Xepon), while the SNP data showed in multiple clades (S11 Fig). Overall, the STR data recapitulates the broad geographical structure of the SNP data but may provide greater resolution of distinct samples at the local scale.

For the *P. vivax* dataset, the PCA analysis (S12 Fig) of both SNP and STR genotypes revealed several distinct clusters that were similar to previous studies [30,31]: South America (Brazil, Colombia, Peru), Mexico, Southeast Asia (Thailand, Vietnam, Myanmar, Cambodia), PNG, Indonesia and Malaysia. The UMAP analyses and neighbor joining trees (S13 Fig) also recapitulates the population structure from the clustering analyses. In the Southeast Asia

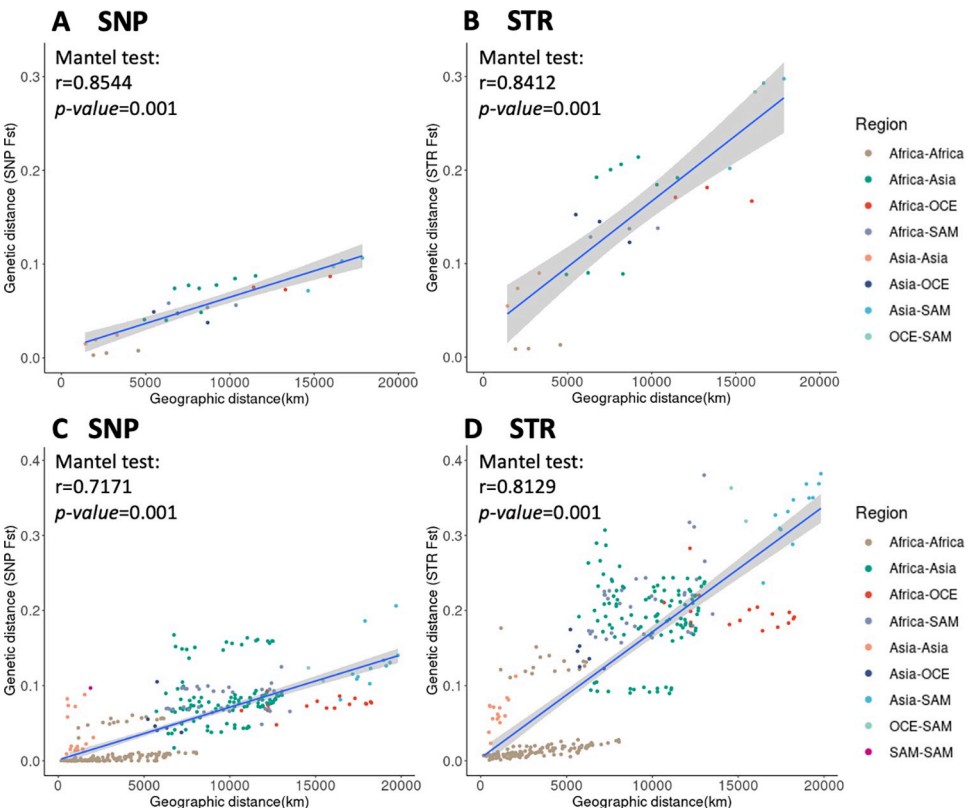

**Fig 4. Pairwise genetic distance ($F_{ST}$) and geographical distances (km) between populations and countries of *P. falciparum*.** (A) SNP data of population pairs. (B) STR data of population pairs. (C) SNP data of country pairs. (D) STR data of country pairs. A Mantel test was used to measure the association.

population, the STR data showed that Myanmar and Vietnam formed their own sub-cluster, while the SNP data showed that they are spread across single- and multiple- country clades (S14 Fig).

**Genome-wide genetic differentiation.** Estimates of population and country differentiation in the *P. falciparum* and *P. vivax* dataset calculated using the STR data were highly correlated with those calculated using the SNP data for both *Jost's D* and $F_{ST}$ (S15 and S16 Figs). Similar to some previous studies [41–43], we found *Jost's D* and $F_{ST}$ tended to produce values higher in magnitude with STRs than with SNPs. The bi-allelic SNP loci limit the information content per locus compared to the more polymorphic STR markers, which have higher allelic diversity per locus and therefore result in higher estimates of *Jost's D* and $F_{ST}$.

There were significant associations between geographic and genetic distances at the population and country level for both SNP and STR data (*P. falciparum*: Mantel test based on pairwise $F_{ST}$, Fig 4; Mantel test based on pairwise *Jost's D*, S17 Fig; *P. vivax*: *Jost's D* and $F_{ST}$, S18 Fig), indicating that genetic differentiation in populations might be the result of isolation by geographic distance. For the *P. falciparum* dataset, we observed that the genetic differentiation between SAM and ESEA is the largest for both SNP and STR data (genome-wide average SNP $F_{ST}$ 0.11, STR $F_{ST}$ 0.30), and that this geographic distance is also the largest among the populations for both SNP and STR $F_{ST}$. It is worth noting that the genetic differentiation was much larger within the Asia region (SAS, ESEA, WSEA) than within the Africa region (CAF, WAF, EAF), despite the geographic distances within Africa is much larger. This may be due to the higher transmission intensity within Africa [44]. For the country level of the *P. falciparum*

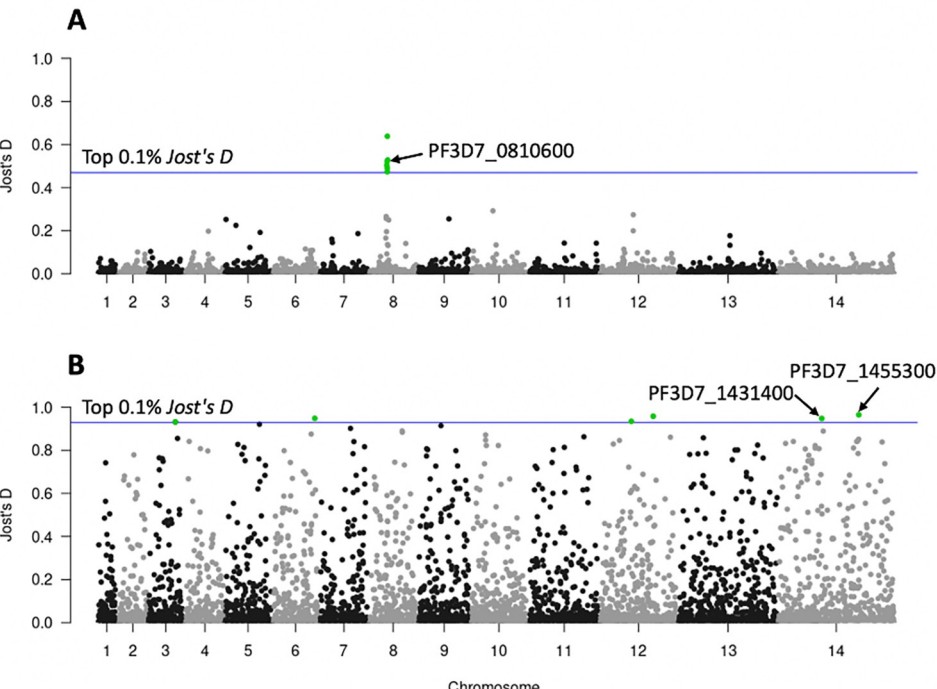

**Fig 5. Genome scans for differentiation, as measured by *Jost's D* values.** (A) CAF, EAF, and WAF samples. (B) Africa region (CAF, EAF, WAF) with Southeast Asia (WSEA, ESEA) region samples. The x-axis represents the chromosomes and the y-axis represents the *Jost's D* values. Each point represents one STR locus with a total 6,768 STRs represented. The blue horizontal line represents the threshold based on the top 0.1% *Jost's D* values.

dataset, within the Africa region, we identified some country pairs that had higher genetic differentiation both in SNP and STR data, this being driven by the Ethiopian genetic differences, which is consistent with the previous studies demonstrating that Ethiopia is a distinct sub-population [21,45].

**Selection signatures related to geographic differentiation.** We performed genome-wide scans of the pairwise *Jost's D* values to identify regions that were differentiated between the populations or the countries in the *P. falciparum* and *P. vivax* dataset. Several genomics regions with high *Jost's D* values were detected at both the global level (different regions) and the local level (different countries). A summary of these comparisons between the *P. falciparum* populations is shown in S6 Table, and the comparison between the *P. vivax* countries is shown in S7 Table.

At the global level in the *P. falciparum* dataset, we found three STRs located in coding regions (within PF3D7_0810600, PF3D7_0810900, PF3D7_0811200) which were highly differentiated between the CAF, EAF, and WAF populations. All three STRs were located on chromosome 8, 0.59–12.3 kb from the drug resistance gene *Pfdhps* (PF3D7_0810800, hydroxymethyldihydropterin pyrophosphokinase-dihydropteroate synthase) (Fig 5A). The first of these STRs (*Jost's D* = 0.52), located within PF3D7_0810600 (chromosome 8 544,455–544,481 kb) is composed of an 'AAT' motif (Fig 5A). 78% samples in CAF and 77.99% samples in WAF have the same genotype as the Pf3D7 reference genome of nine 'AAT' motifs, whereas 86.23% samples in EAF have two 'AAT' insertions. The nonsynonymous mutation (Pf3D7_08_v3:g.543210G>T) in PF3D7_0810600 were detected in artemisinin-resistant cell lines by Rocamora et al. that might play a role in gene expression regulation and subsequently contribute to the artemisinin resistance phenotypes [46].

The second of the three coding STRs is located between the genome reference coordinates 2,260,430–2,260,449 kb, within PF3D7_1455300. It consists of an 'AAT' motif, which was highly differentiated between the Africa region (CAF, EAF, WAF) and the Southeast Asia region (WSEA, ESEA) (Fig 5B). 81.02% of samples in the Africa region have the Pf3D7 reference genotype of seven 'AAT' motifs, whereas 97.17% of samples in the Southeast Asia region have two 'AAT' deletions. PF3D7_1455300 is a conserved *Plasmodium* protein that plays a role in DNA mismatch repair. According to previous work [47], it is a candidate molecular marker for altered DNA repair capability. SNP mutations previously found in this gene may be associated with the phenotype of accelerated resistance to multiple drugs (ARMD). Also, the SNP mutations identified in PF3D7_1455300 (Pf3D7_14_v3:g.2260945T>G) by Xiong et al. have a high frequency in the Southeast Asia population, but cannot be found in African populations, which may be due to selection and thus be a signature for the Southeast Asia population selection signal [47]. In our study, we also found that the STR mutations in PF3D7_1455300 are significantly different in Southeast Asia and Africa. Aside from this locus, another STR within PF3D7_1431400 (surface-related antigen) is located between the genome reference coordinates 1,234,853–1,234,865 kb and consists of a monomer 'T' motif, which also showed high differentiation (Fig 5B). 97.76% samples in Africa have the Pf3D7 reference genotype of 13 'T' motif repeats, whereas 89.52% samples in Southeast Asia have an insertion of 24 'T' repeats.

At the local level in the *P. falciparum* dataset, we also found several STRs which showed differentiation between countries. Within WAF countries, we identified a set of STRs located in coding regions of the genome, with potentially direct functional impact. These STRs appeared to be under positive directional selection: PF3D7_0627800 (acetyl-CoA synthetase), which was predicted as being under balancing selection [48]; and PF3D7_0826100 (HECT-like E3 ubiquitin ligase), found to be possibly involved in resistance mechanism of pyrimethamine [49,50] and which may also be involved in reduced susceptibility to quinine and quinidine [51]. Additionally we also identified STRs in the coding regions of PF3D7_0416000 (RNA-binding protein); PF3D7_0811000 (cullin-1); PF3D7_0811200 (ER membrane protein complex subunit 1); PF3D7_1210400 (general transcription factor 3C polypeptide 5), PF3D7_1409100 (aldo-keto reductase) and two conserved proteins with unknown function: PF3D7_0107100 and PF3D7_0604000. Within EAF countries, selection signals included STRs in PF3D7_0628100 (HECT domain-containing protein 1), which was previously observed to have a strong signature of deviation from neutrality in Gambia based on STR analysis [52]; PF3D7_0527900 (ATP-dependent RNA helicase DDX41); PF3D7_1212900 (bromodomain protein 2); PF3D7_1331100 (DNA polymerase theta); and three conserved proteins with unknown function: PF3D7_0526600, PF3D7_0810900 (0.59kb away from the drug resistance gene *Pfdhps*) and PF3D7_1448500. Two STRs within PF3D7_1433400 (PHD finger protein PHD2) and PF3D7_0926600 (conserved *Plasmodium* membrane protein, unknown function) were highly differentiated between WSEA countries. Several STRs located in the coding region were also found to be highly differentiated between ESEA countries. PF3D7_1225100 (isoleucine—tRNA ligase); PF3D7_0826100 (HECT-like E3 ubiquitin ligase); PF3D7_1317900 (nucleolar complex protein 4); and two conserved protein with unknown function: PF3D7_1233200 and PF3D7_1303800.

We selected the top ten most highly differentiated STRs from each pairwise population and country comparison to evaluate if a small set of highly differentiated STRs could represent population structure. The minimum spanning network can distinguish between two distinct groups of samples (*P. falciparum*: Fig 6; *P. vivax*: S19 Fig). The top ten most highly differentiated STRs from each pairwise comparison of the *P. falciparum* and *P. vivax* dataset have been made available through the R shiny PlasmoSTR, accessible at https://github.com/bahlolab/PlasmoSTR. From the analyses, we can identify STR mutations that are fixed in one population

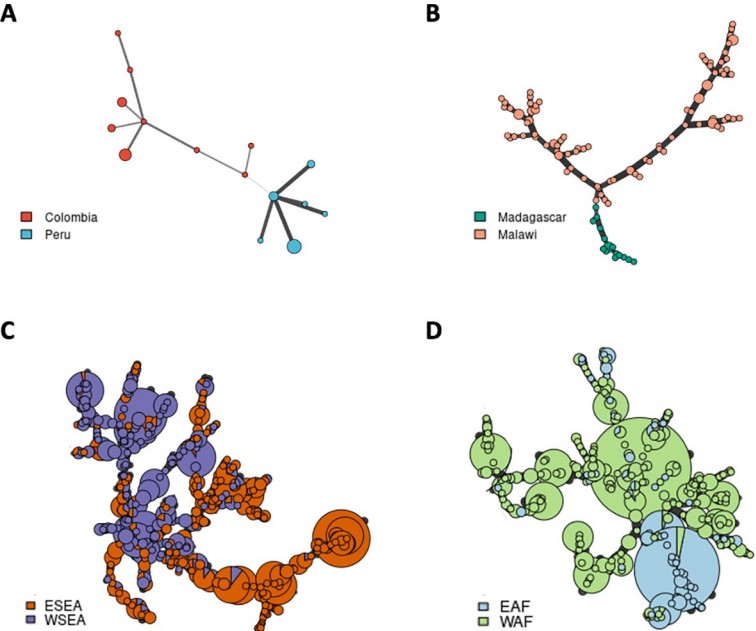

**Fig 6. Minimum spanning network among two groups of *P. falciparum* isolates.** Minimum spanning network using Bruvo's distances based on the ten most informative STR markers. (A) Colombia and Peru from the SAM population. (B) Madagascar and Malawi from the EAF population. (C) ESEA and WSEA populations. (D) EAF and WAF populations. Colors correspond to the country or population. Node sizes correspond to the number of samples. Edge lengths are arbitrary.

but that are distinct from other populations and which may become a signature for the specific population.

**Selection signatures related to drug resistance.** Resistance of malaria parasites to chloroquine is known to be associated with the parasite protein *Pfcrt*. Samples were classified as chloroquine-resistant if they carried the *Pfcrt* 76T allele [21]. Chloroquine-resistance was found in almost all samples from SAM, OCE, SAS, WSEA, and ESEA. It was also found across the Africa region (WAF, CAF, EAF), but the frequency is low, especially in EAF. However, it is noteworthy that all samples from Ethiopia were classified as chloroquine-resistant, and also displayed a higher genetic differentiation with both SNP and STR data with other EAF countries. To identify regions with signatures of selection that may be associated with chloroquine resistance, we calculated the *Jost's D* per STR genome-wide among the 57 EAF drug-resistant and 269 EAF drug-sensitive samples. Average genome-wide *Jost's D* estimates were 0.011, considering the top 0.1% *Jost's D* threshold, signatures of selection were detected at six STR loci that had *Jost's D* values > 0.36, located on chromosomes 7, 11, and 14 (Fig 7). Three STRs were located on chromosome 7, 1.7–22.2 kb from the drug resistance associated gene *Pfcrt* (PF3D7_0709000, chloroquine resistance transporter), demonstrating that drug selection produces chromosomal segments of selective sweeps as we have previously demonstrated with SNP data [53]. One of these STRs (*Jost's D* = 0.82) was located in the genomic promoter region between the genome reference coordinates 392,230–392,268 kb which consists of an 'AT' motif. Interestingly, all of the drug-resistant samples in Ethiopia have one 'AT' deletion compared to the reference genome (S20A Fig), where the EAF drug-sensitive samples range from the six 'AT' deletion to 11 'AT' insertion (S20B Fig). The length of the promoter region is gene-specific, and the levels of gene expression can be increased or decreased by expanding and contracting in length [54]. This is known as expression STRs (eSTRs). To demonstrate

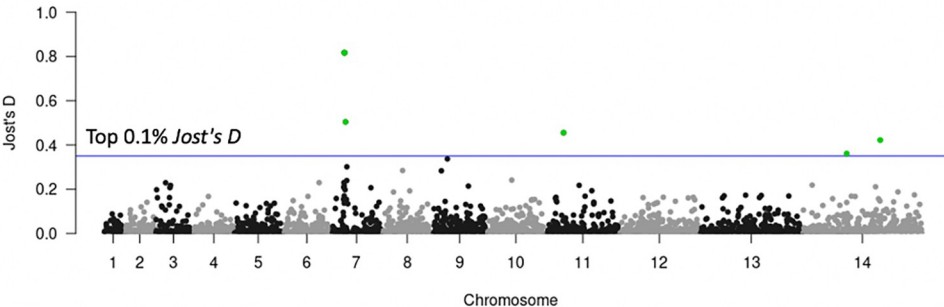

**Fig 7. Genome scans for differentiation for chloroquine drug resistance using 57 EAF drug-resistant and 269 EAF drug-sensitive samples.** The x-axis represents the chromosomes, and the y-axis represents the *Jost's D* values. Each point represents one STR locus. The blue horizontal line represents the top 0.1% *Jost's D* values.

this is a true eSTRs would require a dual WGS, RNA sequencing (RNA-seq) dataset for the parasite, which, to our knowledge, is currently not available, or alternatively a lab-based investigation of expression levels for the different STR genotypes.

## Discussion

In this study we performed STR genotyping on the *P. falciparum* and *P. vivax* genomes comprising more than 3,000 *P. falciparum* and 174 *P. vivax* WGS samples, obtained from across the world. To our knowledge this is the first time this has been attempted for this population dataset and a dataset of this size. The genome-wide characterization and analysis of STR are particularly useful for understanding the genetic diversity, environmental and evolutionary adaptions in *Plasmodium*. We generated a genome-wide catalog of STR variation in *P. falciparum* and *P. vivax* cohort, which provides an important resource to study the STR genetic diversity in different *Plasmodium* populations. Through the extensive comparisons of genome-wide STRs with SNPs in *Plasmodium* population genetics, we demonstrated that STRs are informative and perform better than SNP for accurately determining the local scale population structure. Moreover, we identified several STRs showing significant signals of population differentiation or association with local differences or different antimalarial drug use, suggesting that STR variations may influence the fitness of parasites.

STRs are not routinely analyzed across the whole genome with short read sequencing data because they are difficult to identify and genotype accurately, requiring rigorous QC. In order to attain reliable sets of STRs, some studies have sequenced each sample twice as a technical replicate and used multiple STR calling algorithms to test variant calling accuracy and keep the high-quality STRs [3,55]. However, it is not practical or even possible to use these filtering approaches in large-scale field samples where there are unlikely to be technical replicates. Furthermore, technical replicates are only able to identify a limited set of problematic STRs. To overcome these limitations in the current study, we developed a novel method for quality control of STR genotyping data based on gold standard SNP genotyping data from the same cohort. We demonstrated that this was a successful approach and replicated it in a second *Plasmodium* species, *P. vivax*, demonstrating that this is a method which can be broadly applied to many other species. Our results provided new insights for further exploration of STRs across the whole genome.

We demonstrate that STRs contain additional genetic information which may capture very recent evolution more robustly and could be used to distinguish closely related samples in clonal outbreaks as well as potentially being used to distinguish whether the recurrent

infection represents reinfection or recrudescence. The ability to do so is an important tool for countries aiming for the elimination of *P. falciparum* or *P. vivax* malaria and is particularly important for *P. vivax* with its ability to reactivate malaria from its dormant liver stage. Previous studies used a set of STR markers to identify relationships between primary infection and recurrence-relapse isolates in *P. falciparum* and *P. vivax* [56–58]. Due to the high degree of polymorphism and higher mutation rates, STRs result in higher estimates of genetic distance, which are more likely to provide the estimates of differentiation even in closely related populations. Therefore, STRs are useful for characterization of malaria transmission and inferring population patterns, which can provide complementary information and improve upon SNP-based results, especially in the context of closely related clones such as those observed in local settings or in settings where elimination is close, and the effective population size is small. A previous study identified *de novo* mutations (SNPs and Indels), which can distinguish nominally identical parasites from clonal lineages [59]. STRs can also be used in these applications and their high mutability could add critical further genetic differentiation to differentiate the clonal lineages.

While STRs are generally useful, they have several weaknesses. Previously STR genotyping was limited to small sets of STRs when they were genotyped with low-throughput lab-based methods which is less robust for making inferences about genetic structure. In addition, the complex mechanism of mutations leads to difficulty in calling genotypes, resulting in a higher error rate when compared to SNPs. Overall, our demonstration that STRs can be confidently called in both *P. falciparum* and *P. vivax* WGS field samples enable such future analyses. Concerted genome-wide analyses of both STR and SNP data will provide further insights into the biology and adaptive evolution of *Plasmodium*.

We identified several STRs showing significant signals of genetic differentiation that might occur due to varying antimalarial drug use or local differences. Some STR loci around known drug resistance gene showed significant divergence, including the chloroquine resistance transporter, *Pfcrt* (PF3D7_0709000), and dihydropteroate synthase, *Pfdhps* (PF3D7_0810800, hydroxymethyldihydropterin pyrophosphokinase-dihydropteroate synthase), which are likely due to selective sweeps. Previous studies revealed the STR markers linked with known drug resistance gene demonstrated strong directional selective sweeps and show extensive linkage disequilibrium (LD) surrounding the known drug resistance gene, e.g., *Pfcrt* [60], *Pfdhfr* and *Pfdhps* [61,62]. We also identified several selection signals including the ATP-dependent RNA helicase *Pfdbp1* (PF3D7_0810600), which may be associated with artemisinin resistance [46]; the HECT-like E3 ubiquitin ligase *Pfheul* (PF3D7_0826100), found to be possibly involved in resistance mechanism of pyrimethamine [49,50] and may also be involved in reduced susceptibility to quinine and quinidine [51]; and a conserved *Plasmodium* protein PF3D7_1455300 that may be associated with the phenotype of accelerated resistance to multiple drugs (ARMD) [47]. The STR variations observed in *P. falciparum* drug-resistant samples may reflect the differences in the historical use of antimalarial drugs of different countries and may contribute to the development of local malaria treatment guidelines. Multiple STR loci that had strong signatures of deviation from neutrality were also detected, which was consistent with previous studies [48,52]. However, most of the previous studies detected the selection signatures through association with SNP-based signals, whereas in our study we found that the STR mutations within different genes also play an important role. The key question is whether some of these STRs may actually be the driver mechanism underpinning the selection signals rather than merely showing association due to LD. The small changes in the length of STR mutations may alter protein activity, protein folding efficiency, stability, or aggregation [9], and the levels of gene expression can be increased or decreased by expanding and contracting in length that allows the parasite to adapt under selective pressure [54]. These signals could be

actively pursued in laboratories to investigate whether the STR signals directly affect relevant expression signals as eSTRs. Unlike the human Genotype–Tissue Expression Project (GTEx) [63], a comprehensive public resource, including both WGS and RNA-seq datasets, which can identify STRs associated with expression of nearby genes is not available. Some previous studies used joint genome and transcriptome data to evaluate the impact of STR variation on gene expression in humans and demonstrated that eSTRs contribute to a range of human phenotypes [64,65]. One study in plants also identified a link between STR length variation and gene expression variation in *Arabidopsis thaliana* [66]. These findings among different species highlight the importance of STR variation on gene expression. Further work could perform genome-wide *Plasmodium* eSTRs detection which will help to understand the full functional impact of STR variation. The catalogue of STRs developed in this paper is an important contribution towards this future endeavour.

One limitation in our study is that we are unable to determine if the STRs we detected as being associated with selection signals are true eSTRs due to the inability to assess expression data for the *Plasmodium* samples analyzed. Many novel candidate genomic regions that were likely under recent positive directional selection were also detected in our study, possibly revealing recent signals of selection not yet observable with SNP markers. The second limitation is that the short-read WGS data cannot span large STR alleles, which means that our set of STRs is still incomplete. The use of PCR-free sequencing libraries and long-read WGS data is expected to improve STR genotyping accuracy [59] and will capture more complete sets of STRs, which will provide new insights into subtelomeric regions that contain multigene families involved in immune evasion which we were unable to explore in this work.

## Conclusions

In this paper, we report the first large-scale *in-silico* STR study performed in more than 3,000 *P. falciparum* and 174 *P. vivax* WGS worldwide samples. We developed a novel method for quality control of STR genotyping data based on gold standard SNP genotyping data, which provides new insights for further exploration of STRs across the whole genome. Furthermore, a set of genome-wide high-quality STRs were then used to study parasite population genetics and compared to genome-wide SNP genotyping data, revealing both high consistency with SNP based signals, as well as identifying some signals unique to the STR marker data. These results demonstrate that the identification of highly informative STR markers from large population screening is a powerful approach to study the genetic diversity, population structures and genomic signatures of selection on *P. falciparum* and *P. vivax*. In addition, the genome-wide information about genetic variation and other characteristics of STRs in *Plasmodium* have been available in an interactive web-based R Shiny application PlasmoSTR (https://github.com/bahlolab/PlasmoSTR).

## Materials and methods

### Data

**MalariaGEN global *P. falciparum* dataset.** All samples and metadata were obtained through the MalariaGEN *Plasmodium falciparum* Community Project (https://www.malariagen.net/resource/26) [21]. We retrieved the data in fastq file format from the Sequence Read Archive (SRA). The *P. falciparum* dataset consists of 3,241 monoclonal (within-host infection fixation index $F_{ws} > 0.95$ downloaded from MalariaGEN) samples. Metadata was available in the form of population labels for all samples representing the country of origin of each sample at both a population (8 levels) and country level (27 levels). The populations were SAM, WAF, CAF, EAF, SAS, WSEA, ESEA and OCE, and the countries were: Ghana,

Cambodia, Bangladesh, Thailand, Colombia, Malawi, Guinea, Uganda, Ethiopia, Mali, Senegal, Gambia, Mauritania, Peru, Nigeria, Myanmar, Laos, Vietnam, Kenya, Tanzania, Papua New Guinea, Burkina Faso, Congo DR, Madagascar, Cameroon, Ivory Coast, and Benin [21]. On the basis of previously published genetic markers including SNPs and copy number variations (CNVs), all samples are classified into different types of drug resistance [21]. As previously described [21], sequencing was performed using Illumina HiSeq 2000 paired-end sequencing platform. The *P. falciparum* Pf3D7 (v3 PlasmoDB-41) was used as the reference genome and was downloaded from PlasmoDB [67]. In this study, we only considered monoclonal samples because STR genotyping algorithms such as HipSTR are optimised for diploid and haploid chromosomes, without considering the possibility of multiplicity of infection (MOI) [24,26].

**Global *P. vivax* dataset.**   The dataset for *P.vivax* comprised 353 previously published samples in *Plasmodium vivax* Genome Variation project (https://www.malariagen.net/projects/p-vivax-genome-variation) as described in Pearson et al. [31] and data from Hupalo et al. [30], which are sampled from multiple countries around the world. Fastq files were also downloaded from SRA. The whole genome sequencing was performed using Illumina-based sequencing platforms. The *P. vivax* genome PvP01 (PlasmoDB release 41) was used as the reference genome and was downloaded from PlasmoDB [67].

## Methods

**SNP genotyping.**   SNPs and deletions/insertions (Indels) were called using the standard best practice from Genome Analysis Toolkit (GATK) version 4.0.12.0 implemented in nextflow [20,68] (https://github.com/gatk-workflows/gatk4-germline-snps-indels). The pipeline generates a final joint VCF file for all samples. Variants were further removed with the following filtering thresholds: Quality of Depth (QD) < 20, Mapping Quality (MQ) < 50, MQ Rank Sum (MQRankSum) < -2, Strand Odds Ratio (SOR) > 1, and Read Position Rank Sum (ReadPosRankSum) less than -4 or greater than 4. SnpEff was used to annotate variants based on the *P. falciparum* Pf3D7 and *P. vivax* PvP01 reference genome [69]. SNPs were excluded if they were: (i) indels, (ii) not biallelic, (iii) variants in genes from the surface antigen (VSA) [70] families, (iv) not in core genome region defined by Miles et al. [22] for *P. falciparum* and Pearson et al. [31] for *P. vivax*, (v) if their minor allele frequency (MAF) was less than 1% in all populations, or (vi) their missing genotype frequency was higher than 10%.

**STR genotyping.**   We initially identified the composition and distribution of STRs in the *P. falciparum* Pf3D7 and *P. vivax* PvP01 reference genome using Tandem Repeats Finder (TRF Version 4.09) [34]. The parameters used for TRF were: the alignment weights for matching (Match) equal to 2, mismatching (Mismatch) penalty is 7, indel (Delta) penalty is 7, the match probability (PM) is 80, the indel probability (PI) is 10, the minimum alignment score (Minscore) is 20, the maximum period size (MaxPeriod) (the pattern size of the tandem repeat) to report is 500bp. Additional post-processing steps of the TRF output files were performed by removing STRs: (i) with overlapping STRs, (ii) with motif period size > 9bp, (iii) repeat number of the motif < 3, (iv) where the percent of matches ≤ 85%, (v) where the percentage of indels ≥ 5%, (vi) the repeat length was larger than 70bp, as the genotype call rate declined for longer tandem repeats. Genome-wide STR genotyping was performed with HipSTR (Version 0.6.2) [26] using the haploid version under the default parameters. STRs were then excluded if they were from VSA [70] families, or if they were not in the core genome [22,31], or if their missing genotype rate was higher than 10%. The VariantAnnotation [71] R package (Version 1.32.0) was used to annotate variants making use of the Pf3D7 and PvP01 gene annotation in GFF format.

To evaluate the performance of HipSTR's STR genotyping, we used the MalariaGEN Pf3k genetic cross dataset to assess the genotypic errors (https://www.malariagen.net/data/pf3K-5) [22]. This dataset contains six parents and 92 progenies derived from three experimental genetic crosses of *P. falciparum*. The 3D7×HB3 cross comprised both parents and 19 progenies [35], the HB3×Dd2 cross comprised both parents and 35 progenies [36], and the 7G8×GB4 cross comprised both parents and 38 progenies [37]. We compared genotypes observed in the parents and the progeny to detect Mendelian errors (MEs, the progeny has an allele found in neither of the parents). MER is defined as the number of MEs divided by the total number of informative variants in each genetic cross data. We also compared the genotype discordance between biological replicates in this dataset. The 3D7×HB3 cross contains one replicate for clone C01 and three replicates for clone C02 (six replicate pairs), a total of seven replicate pairs. The 7G8×GB4 cross contains one replicate for 10 progeny clones, a total of 10 replicate pairs. We then computed genotype discordance for each replicate pair. The genotype discordance for each replicate pair is defined as the number of variants with a discordant genotype call divided by the total number of informative variants in each genetic cross data. In addition, we compared the MER and genotype discordance between biological replicates of SNP genotyping and STR genotyping data generated from the same dataset. The MER and genotype discordance of SNP and STR genotyping was determined separately within each genetic cross. These are summarised in S8 and S9 Tables. We then checked the genotype accuracy of 20,196 STRs (6,768 high-quality STRs and 13,428 low-quality based on model) that we genotyped in 3,047 *P. falciparum* samples in the same dataset (Pf3k genetic cross dataset) by applying the same filtering strategy as 3,047 *P. falciparum* samples. A large proportion of the STRs were found to be non-informative (invariant) in the Pf3k genetic cross dataset (16,966/20,196), and as such, we only compared the informative STRs. These are summarised in S10 and S11 Tables. Next, we removed the SNPs and STRs with MEs and genotype discordance between biological replicates, retaining 9,937 STRs and 25,225 SNPs from 94 samples (excluding four samples with more than 10% missing genotypes). We then assessed the number of segregating sites (the number of polymorphic positions in the genotype data) for both STRs and SNPs between pairwise samples. To further assess HipSTR's genotyping accuracy, we also used 10 *P. falciparum* STR markers proposed by Anderson et al. (1999) [72]. These markers have been genotyped with GE on Applied Biosystems 3700 (ABI3700) in a set of in-house *P. falciparum* samples. Whole-genome sequencing was also performed on these samples and included in MalariaGEN *Plasmodium falciparum* Community Project (https://www.malariagen.net/resource/26) [21]. GE is typically taken to be the gold standard for STR genotyping. GE data was only available for Milne Bay and East Sepik samples (90 samples). S12 Table represents the set of markers we have for *P. falciparum* in addition to how many samples had genotypes called by HipSTR at these specific markers. The locations of the markers were obtained by using a BLAST search with PlasmoDB [67] on the primer sequences for each STR marker which were presented in Anderson et al. (1999) [72]; Figan et al. (2018) [15]; Greenhouse et al. (2006) [73]. Given that the locations of the markers are known, we can compare HipSTR's genotype calls to the length of the STR markers as determined by the GE procedure. It is important to note that the reported length of STRs from GE are generally shifted by a fixed number of base pairs due to the primers being used, which add to the product length [72]. Regardless, there should still be a linear relationship between the two classifications if HipSTR is predicting genotypes well.

**Characterization of within-host diversity.**   We applied the $F_{ws}$ metric to the *P. falciparum* and *P. vivax* dataset to determine samples that had multiple infections [32]. Samples with $F_{ws}$ < 0.95 were considered multiple infections [33]. $F_{ws}$ was calculated using the moimix (Version

0.0.2.9001) [74] R package. Samples with multiple infections were excluded from further analysis.

**Multivariable logistic regression modeling for measurement and prediction of the quality of STRs.** Although *in-silico* STR genotyping methods have QC metrics that can be used to identify well performing STRs, these have been shown to retain many poorly performing STRs, which are not easy to identify. In order to collate a set of high-quality STRs, we developed a complex filtering strategy based on leveraging genetic distance between samples as determined by SNPs and STRs, aiming to identify further, more precise STR relevant metrics that could be applied to identify high quality STRs. The rationale here is that variants with high genotyping accuracy should more accurately represent the population structure of field samples. Based on this, we developed a SNP PCA based approach to capture the signal STRs. PCA was first performed to investigate potential population structure using the SNP genotype data (*P. falciparum*: 213,757 SNPs loci; *P. vivax*: 188,571 SNPs loci). The top ten principal components were chosen. The squared Spearman's correlation coefficient ($R^2$) was then used to assess correlations between the top ten SNP PCs values and each STR by using the repeat units (number of times the motif is repeated in tandem) for each sample, using only those STRs retained after the initial QC step described above (*P. falciparum*: 20,196 STRs; *P. vivax*: 23,146 STRs). STRs that correlated with an $R^2$ above a permutation derived threshold with any one of the ten significant SNP PCs were deemed to be high quality STRs. The optimal cut-point of the correlation to distinguish high-quality or low-quality STRs was determined using the resampling permutation test where the population labels were permuted between samples to derive a null distribution to determine an appropriate correlation threshold that maximised the difference between high and low quality STRs. This was determined using the ROC AUC.

A set of metrics including QC metrics from HipSTR and metrics that were deemed useful were derived and used for the prediction of the STR quality. These are summarised in S2 Table. The Z-score standardization method was used to normalize these metrics. Considering the large difference of sample size between the *P. falciparum* and *P. vivax* dataset, when calculating the metrics that are associated with population structure, the *P. falciparum* dataset is based on 8 population-level labels and the *P. vivax* dataset is based on 11 country-level labels. A multivariable logistic regression analysis was then performed to identify potential predictors of STR quality. Empirical clustering and subsequent labelling was performed by using SNP genotype data to calculate the identity-by-state (IBS) pairwise distance between samples using the SNPRelate (Version 1.20.1) [75] R package. Clustering analysis was then performed to assign the samples clusters which were assumed to represent geographical regions.

Model selection in the multivariable regression models was employed for stepwise regression analysis based on the Akaike information criterion (AIC) using the R package MASS (Version 7.3–51.5). The effectiveness of the prediction was evaluated by calculating the AUC on the ROC curve. To assess the predictive performance of the logistic regression model, the large *P. falciparum* dataset was randomly separated into five combinations of training and test sets of 80/20 split, and fivefold cross-validation was performed. Predictive performance was measured with the AUC in the testing model. To assess the robustness of the performance of the model with respect to different size datasets, the large *P. falciparum* dataset was also randomly separated into five combinations of training and test sets of each 70/20, 60/20, 50/20, 40/20, 30/20, 20/20, and 10/20 splits, and fivefold cross-validation was performed. To select the optimal cut-off value to remove low-quality STRs, the predicted probabilities are sorted into five bins ([0, 0.2], [0.2, 0.4], [0.4, 0.6], [0.6, 0.8], [0.8, 1]). For each bin we randomly selected 500 STRs and calculated the correlation of sample pairwise distances of STR and SNP based on PCA analysis, and repeated this 100 times to calculate the mean value of correlation.

The R script used to perform the multivariable logistic regression modeling for measurement and prediction of the quality of STRs is available on https://github.com/bahlolab/PlasmoSTR.

**Population structure analysis.** To investigate the major geographical division of population structure that could be determined with the final set of STR markers, the SNP-based and STR-based PCA of all 3,047 *P. falciparum* and 174 *P. vivax* samples were performed separately. In the SNP-based PCA we used 213,757 SNPs for *P. falciparum* and 188,571 SNPs for *P. vivax*. In the STR-based PCA, we used 6,768 (2,563 mononucleotide STR and 4,205 polynucleotide STR) high-quality loci for *P. falciparum* and 3,496 (1,648 mononucleotide STR and 1,848 polynucleotide STR) for *P. vivax* selected by the logistic regression model based on predicted probabilities. PCA plots were constructed from the analysis. UMAP [76] was performed after selecting the significant PCs using the umap (Version 0.2.4.1) R package. SNP and high-quality STR loci across the whole genome were used to calculate the average IBS distances as the average genetic dissimilarity between pairs of individuals. The R package SNPRelate [75] was used to calculate the IBS values for SNP data and a method based on Bruvo's distance [77] was used for STR data, providing a stepwise mutation model appropriate for microsatellite markers. Neighbor joining trees were then produced using the R package ape (Version 5.4–1) [78] and ggtree (Version 2.0.4) [79].

**Genome-wide genetic differentiation.** Pairwise estimates of genetic differentiation ($F_{ST}$) between all pairs of populations and countries defined by geographic origin were calculated using the R package SNPRelate [75] for SNP data and the R package hierfstat (Version 0.5–7) [80] for microsatellites based on the method of Weir and Cockerham (1984) [81]. The degree of population differentiation was also measured by calculating *Jost's D* using the R package mmod (Version 1.3.3) [82], which is a superior diversity measure for highly polymorphic loci proposed by Jost [83]. The geographic distance between different populations (based on great circle distances using the haversine [84]) and countries (km), were calculated using the R packages sf (Version 1.0–3) [85], maps(Version 3.4.0), units (Version 0.7–2) [86], geosphere (Version 1.5–14), and rnaturalearth (Version 0.1.0). The Mantel test was performed by the R packages vegan (Version 2.5–7) [87] to study the correlations between pairwise values of genetic distance and geographical distance between populations and countries, and also used to check the correlation between pairwise differentiation measures (*Jost's D* and $F_{ST}$) from SNPs and STRs data.

**Selection signatures related to geographic differentiation.** Global *Jost's D* were calculated per STR for each pairwise population combination (*P. falciparum*: 8 populations = 28 comparisons) and country combination (*P. falciparum*: 26 countries = 325 combinations) using the mmod [82] R package. For *P. vivax*, we only compared the countries with sample sizes larger than 10 (*P. vivax*: 5 countries = 10 combinations) as many countries had few samples ($< 5$). To identify regions with strong signatures of selection, the top 0.1% *Jost's D* values were used to set the threshold to represent a selection signature. Genome-wide distribution of selection signatures was visualized by plotting the *Jost's D* against chromosome positions. To find out if a small set of highly differentiated STRs, such as those routinely used in capillary genotyping based STR analysis, could possibly show the population structure, we extracted the top ten most highly differentiated STRs from each pairwise comparison. The highly differentiated multilocus genotypes (MLGs) from these ten STRs were then used to construct a minimum spanning network (MSN) plot using Bruvo's distance using the R package poppr (Version 2.9.0) [88].

**Selection signatures related to drug resistance.** According to published genetic markers, all *P. falciparum* samples are classified into different types of drug resistance in the MalariaGEN *Plasmodium falciparum* Community Project [21]. *Jost's D* was calculated per STR among

the drug-resistant and drug-sensitive samples using the mmod [82] R package to explore the genetic differentiation. The top 0.1% *Jost's D* values were used to set the threshold to represent a selection signature. Considering that the malaria parasite population genetic structure varies substantially among the different populations due to different malaria control efforts, signatures of selection related to drug resistance were only performed for the comparison within subpopulations.

## Supporting information

**S1 Fig. Evaluation of the Mendelian error rate of SNP and STR genotype accuracy in three *P. falciparum* genetic crosses.** Test results (p-values) are from a two-sample Wilcoxon rank-sum test.
(TIF)

**S2 Fig. Discordance rates of the SNP and STR genotypes calculated from biological replicates in two *P. falciparum* genetic crosses.** Test results (p-values) are from a two-sample Wilcoxon rank-sum test.
(TIF)

**S3 Fig. Evaluation of the genotype accuracy of the high-quality and low-quality STRs defined by the model in three *P. falciparum* genetic crosses dataset.** (A) The Mendelian error rate of STR genotypes. (B) Discordance rate of STR genotypes between biological replicates. Test results (p-values) are from a two-sample Wilcoxon rank-sum test.
(TIF)

**S4 Fig. Comparison of STR and SNP segregating sites.** (A) The number of segregating sites between pairwise samples for both STRs and SNPs. (B) The relative number of segregating sites (segregating sites/No. of variants) between pairwise samples for both STRs and SNPs. The dashed line represents y = x. Pearson correlation coefficient r and test results (p-values) are indicated for each plot.
(TIF)

**S5 Fig. Comparison of HipSTR STR genotype calls with STR calls from gel electrophoresis (GE) data for *P. falciparum*.** Bubble plots representing the GE allele calls are plotted against HipSTR's calls, where the left plot has been shifted such that the bottom leftmost point lies on the origin. In the right plots points are coloured according to the region where samples originated from. The line represents y = x. It is important to note that in most of these plots, the Milne Bay samples are typically off the line y = x, indicating potential underlying issues with the GE calls for Milne Bay samples.
(TIF)

**S6 Fig. ROC curves and AUC values of the *P. falciparum* complete dataset and five training datasets.** (A) The mononucleotide STR model. (B) The polynucleotide STR model.
(TIF)

**S7 Fig. AUC values of the *P. falciparum* fivefold cross validation-datasets (80/20, 70/20, 60/20, 50/20, 40/20, 30/20, 20/20, and 10/20 splits).** (A) The mononucleotide STR model. (B) The polynucleotide STR model.
(TIF)

**S8 Fig. Principal component analysis of the 3,047 *P. falciparum* samples of SNP and STR data.** (A) SNP-based PCA based on 213,757 loci. (B) STR-based PCA based on 6,768 (2,563

mononucleotide STR and 4,205 polynucleotide STR) high-quality loci.
(TIF)

**S9 Fig. Sub-population structure analysis of the SAM and EAF population *P. falciparum* samples of SNP and STR data.** (A) UMAP on the top five principal components (PCs) of the SNP data (SAM countries). (B) UMAP on the top five PCs of the STR data (SAM countries). Colouring the points by the SAM countries. (C) UMAP on the top five PCs of the SNP data (EAF countries). (D) UMAP on the top five PCs of the STR data (EAF countries). Colouring the points by the EAF countries.
(TIF)

**S10 Fig. Population structure analysis of the 81 *P. falciparum* samples of SNP and STR data.** (A) Map showing the geographical location of the three different sample collection sites from PNG. Map drawn with the data from Natural Earth (http://www.naturalearthdata.com/) by the R package rnaturalearth (Version 0.1.0) (https://github.com/ropensci/rnaturalearth) under a CC BY license. (B) Neighbor joining tree based on the SNP data. (C) Neighbor joining tree based on the STR data. Branches are colored according to the site.
(TIF)

**S11 Fig. Population structure analysis of the 79 *P. falciparum* samples of SNP and STR data.** (A) Map showing the geographical location of the two different sample collection sites from Laos. Map drawn with the data from Natural Earth (http://www.naturalearthdata.com/) by the R package rnaturalearth (Version 0.1.0) (https://github.com/ropensci/rnaturalearth) under a CC BY license. (B) Neighbor joining tree based on the SNP data. (C) Neighbor joining tree based on the STR data. Branches are colored according to the site.
(TIF)

**S12 Fig. Principal component analysis of the 174 *P. vivax* samples of SNP and STR data.** (A) SNP-based PCA based on 188,571 loci. (B) STR-based PCA based on 3,496 (1,648 mononucleotide STR and 1,848 polynucleotide STR) high-quality loci.
(TIF)

**S13 Fig. Population structure analysis of the 174 *P. vivax* samples of SNP and STR data.** (A) UMAP clustering of the top five PCs of the SNP data with different colors representing the 11 different countries. (B) UMAP clustering of the top five PCs of the STR data. (C) Neighbor joining tree based on the SNP data. (D) Neighbor joining tree based on the STR data. Branches are colored according to the country.
(TIF)

**S14 Fig. Population structure analysis of the 86 *P. vivax* samples of SNP and STR data.** (A) Map showing the geographical location of the four countries. Map drawn with the data from Natural Earth (http://www.naturalearthdata.com/) by the R package rnaturalearth (Version 0.1.0) (https://github.com/ropensci/rnaturalearth) under a CC BY license. (B) Neighbor joining tree based on the SNP data. (C) Neighbor joining tree based on the STR data. Branches are colored according to the country.
(TIF)

**S15 Fig. A comparison of measures of genetic differentiation (*Jost's D* and $F_{ST}$) estimates using SNP and STR data of *P. falciparum*.** (A) *Jost's D* of population pairs (*Mantel r* = 0.996, *P* = 0.001). (B) *Jost's D* of country pairs (*Mantel r* = 0.996, *P* = 0.001). (C) $F_{ST}$ of population pairs (*Mantel r* = 0.97, *P* = 0.001). (D) $F_{ST}$ of country pairs (*Mantel r* = 0.90, *P* = 0.001). Mantel

tests were used to measure the correlation.
(TIF)

**S16 Fig. A comparison of measures of genetic differentiation (*Jost's D* and $F_{ST}$) estimates using SNP and STR data of *P. vivax*.** (A) *Jost's D* of country pairs (*Mantel r* = 0.9678, *P* = 0.001). (B) $F_{ST}$ of country pairs (*Mantel r* = 0.9185, *P* = 0.001). Mantel tests were used to measure the correlation.
(TIF)

**S17 Fig. Pairwise genetic distance (*Jost's D*) and geographical distances (km) between populations and countries of *P. falciparum*.** (A) SNP data of population pairs. (B) STR data of population pairs. (C) SNP data of country pairs. (D) STR data of country pairs. A Mantel test was used to measure the association.
(TIF)

**S18 Fig. Pairwise genetic distance (*Jost's D* and $F_{ST}$) and geographical distances (km) between countries of *P. vivax*.** (A) SNP data of country pairs (*Jost's D*). (B) STR data of country pairs (*Jost's D*). (C) SNP data of country pairs ($F_{ST}$). (D) STR data of country pairs ($F_{ST}$). A Mantel test was used to measure the association.
(TIF)

**S19 Fig. Minimum spanning network using Bruvo's distances based on the ten most informative STR markers showing the relationship among two groups of *P. vivax* isolates.** (A) Cambodia and Thailand. (B) Colombia and Peru. (C) Mexico and Peru. Colors correspond to the country. Node sizes correspond to the number of samples. Edge lengths are arbitrary.
(TIF)

**S20 Fig. Examples of corresponding genotypes.** (A) One drug-resistant sample. (B) One drug-sensitive sample.
(TIF)

**S1 Table. Comparison between the allele length (bp) of four *P. falciparum* STR markers for HipSTR calls and gel electrophoresis (GE) data.**
(XLSX)

**S2 Table. Baseline variables used for prediction of the STR quality.**
(XLSX)

**S3 Table. Multivariable logistic regression model's estimated coefficients and respective 95% confidence intervals.** The model was fitted on the *P. falciparum* dataset without the population origin label.
(XLSX)

**S4 Table. Multivariable logistic regression model's estimated coefficients and respective 95% confidence intervals.** The model was fitted on the *P. vivax* dataset.
(XLSX)

**S5 Table. Multivariable logistic regression model's estimated coefficients and respective 95% confidence intervals.** The model was fitted on the *P. vivax* dataset without the population origin label.
(XLSX)

**S6 Table. Detected selection signatures (located in the coding region) between the *P. falciparum* populations containing the top 0.1% of STR.**
(XLSX)

**S7 Table. Detected selection signatures (located in the coding region) between the *P. vivax* countries containing the top 0.1% of STR.**
(XLSX)

**S8 Table. The Mendelian error rate of SNP and STR genotypes in three *P. falciparum* genetic crosses.**
(XLSX)

**S9 Table. Discordance rates of SNP and STR genotypes calculated from biological replicates in two *P. falciparum* genetic crosses.**
(XLSX)

**S10 Table. Mendelian error rates of STR genotypes of the high-quality and low-quality STRs defined by the model in the three *P. falciparum* genetic crosses dataset.**
(XLSX)

**S11 Table. Discordance rates of STR genotypes between biological replicates of the high-quality and low-quality STRs defined by the model in two *P. falciparum* genetic crosses dataset.**
(XLSX)

**S12 Table. *P. falciparum* STR markers used in the analysis and the number of samples genotyped at each STR marker.** The symbol "*" denotes markers which were dropped in the TRF post-processing phase rather than the HipSTR phase. The table also highlights the location of the markers on the Pf3D7 reference genome as determined by the BLAST search.
(XLSX)

## Acknowledgments

This publication uses data from the MalariaGEN *Plasmodium falciparum* Community Project as described in 'An open dataset of *Plasmodium falciparum* genome variation in 7,000 worldwide samples. MalariaGEN et al, Wellcome Open Research 2021642 DOI: 10.12688/wellcomeopenres.16168.1' [21], from the MalariaGEN Pf3k genetic cross dataset (https://www.malariagen.net/data/pf3K-5) [22], from the MalariaGEN *P. vivax* Genome Variation project, as described in Pearson et al., *Nature Genetics*, 2016 (https://doi.org/10.1038/ng.3599) [31] and from the Broad Institute, as described in Hupalo et al., *Nature Genetics* 2016 (https://doi.org/10.1038/ng.3588) [30]. We thank the MalariaGEN Consortium and Hupalo et al. [30] for allowing the use of this data.

## Author Contributions

**Conceptualization:** Melanie Bahlo.

**Data curation:** Jiru Han, Jacob E. Munro.

**Formal analysis:** Jiru Han.

**Funding acquisition:** Melanie Bahlo.

**Investigation:** Jiru Han, Jacob E. Munro, Anthony Kocoski, Alyssa E. Barry, Melanie Bahlo.

**Methodology:** Jiru Han, Jacob E. Munro, Melanie Bahlo.

**Project administration:** Melanie Bahlo.

**Resources:** Alyssa E. Barry, Melanie Bahlo.

**Supervision:** Melanie Bahlo.

**Validation:** Jiru Han.

**Visualization:** Jiru Han.

**Writing – original draft:** Jiru Han.

**Writing – review & editing:** Jiru Han, Jacob E. Munro, Anthony Kocoski, Alyssa E. Barry, Melanie Bahlo.

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
