## [Decision Letter · Decision Letter 0]

18 Aug 2021

Dear Dr Bahlo,

Thank you very much for submitting your Research Article entitled 'Population-level genome-wide STR typing in Plasmodium species reveals higher resolution population structure and genetic diversity relative to SNP typing' to PLOS Genetics.

The manuscript was fully evaluated at the editorial level and by independent peer reviewers. The reviewers appreciated the attention to an important problem, but raised some substantial concerns about the current manuscript. Based on the reviews, we will not be able to accept this version of the manuscript, but we would be willing to review a much-revised version. We cannot, of course, promise publication at that time.

If you decide to revise the manuscript for further consideration at PLOS Genetics, please aim to resubmit within the next 60 days, unless it will take extra time to address the concerns of the reviewers, in which case we would appreciate an expected resubmission date by email to plosgenetics@plos.org.

[LINK]

We are sorry that we cannot be more positive about your manuscript at this stage. Please do not hesitate to contact us if you have any concerns or questions.

Yours sincerely,

Giorgio Sirugo

Associate Editor

PLOS Genetics

Hua Tang

Section Editor: Natural Variation

PLOS Genetics

Reviewer's Responses to Questions

**Comments to the Authors:**

Reviewer #1: The manuscript by Han et al, describes tools for exploring and identifying STRs in the genome of P. falciparum and P. vivax. As has been the case for other species, methods to identify and filter SNPs have well outpaced STRs for understand malaria population genetics. To this end the authors propose a regression based approach to aid in identifying genuine variation. While the area of STR typing is of high interest to malaria researchers, there are concerns I have on the novelty, validation and generalizability of the paper. Overall, I wanted to see a greater investment in showing how STRs may be superior to SNPs to understand structure and selection in the parasite genome, especially over timescale where the higher mutability of STRs adds critical detail. STRs could be a really powerful alternative to SNPs, I would love to see this demonstrated more clearly.

Novelty:

The authors state that this is the “first large-scale STR typing study”. I disagree with this characterization. The data in the study is a subset of 7,000 samples published by the MalariaGEN consortium. A paper cited in the current manuscript (Ahoidi et al, Wellcome Open Research) reports ~3,000,000 indels typed across the 7,000 samples. STRs clearly compose a subset of these. In addition, the authors cite several other papers (including some which also use HipSTR) which report significantly sized data (albeit not in the thousands). One omission is the work of Redmond et al (MBE May 2018) where a careful treatment of InDels/STRs is made using PCR-free libraries and far longer reads (250bp) more effectively controlling error rates. Several of the cited works also use crafty validation approaches (inheritance in genetic crosses/augmentation of reference sequences) which could inform the current work.

Some chunk of the paper has been devoted to the distribution of STRs through the genome. To some extent this is a retreading of other papers (notably Hamilton et al and McDew-White et al).

Validation:

The comparison to the gel electrophoresis is a great idea, though oddly no estimate of the accuracy was determined in this manner. The data looks supportive – though it is difficult to visually interpret the point sizes. Can the numbers here be made explicit and used to derived a measure of accuracy

To have this as the sole direct measure of accuracy is disappointing. There are manifold ways to derive this – as above, either using mendelian error rates from genetic crosses, augmenting reference sequences, or looking at concordance between technical replicates would all be valuable. Technical reps are gathered, it is not clear how these have been leveraged.

I have significant issues with the regression based approach. To my reading, STR quality was determined by how well an STRs genotype conformed to an estimate of population structure derived from SNPs (either the first 5 or 10 PCs, depending on the text vs. methods). STR quality is not really defined clearly, though I have a fundamental issue (or a fundamental misunderstanding) of how this approach works. Firstly, I would not expect any specific STR to match the global population structure from SNPs. The STRs reflect the local structure at that point along the chromosome. The local structure would not necessarily reflect the global (for instance in the case of selective sweeps surrounding drug resistance genes), and even locally the mutability of STRs would suggest that the impact of sweeps would be retained over different timescales than for SNPs so that local structure would not necessarily be maintained.

The model is split by mono and polynucleotide repeats. However, particularly for falciparum, there are almost no 2bp repeats in coding regions, and almost no 3bp repeats outside of them. Sequence quality varies considerably between coding and non-coding regions. How is this accounted for?

A goal of the paper is to capture differences between SNPs and STRs for population structure, if there is a filtering of STRs to only retain those which reflect structure from SNPs then there seems little value in using them instead of SNPs. The regression model appears to be validated based upon how closely the STRs match the population structure. It would appear to be trained to find a set of markers which match the population structure defined by SNPs rather than to find accurately called STRs.

I also find it odd that expected heterozygosity and Jost’s D are in the model. Surely, these should be independent of the model so that signal of selection are not driven by population structure?

Selection:

The scans for selection for drug resistance genes lacks a clear rationale. EAF samples are split by the presence of K76T at pfcrt. Correspondingly, divergence around this allele is found, presumably due to the global selective sweep at this locus. Why would any other loci show significant divergence? Chloroquine has not been used for some time in east Africa and any selection maintaining interchromosomal linkage is likely diminished.

For the analysis in SE Asia, this is again quite confusing. Artemisinin resistance has arisen on hundreds of different genetic backgrounds, it is very challenging to understand what these other loci mean. Notably, they appear all reflect differences in historical drug use between Laos and Thailand, rather than being involved in artemisinin resistance. For instance, the chr 5 is near pfmdr1, the chr 12 locus is near GTP-cyclohydrolase suggesting the impact of mefloquine/anti-folate use.

Were attempts made to use alternative selection statistics, for instance those which use haplotype information. A highly mutable marker would seem to offer a very low background signal for such approaches.

For figure 4 each y-axis is to a different height making it difficult to compare across plots.

I was also confused why FWS filtering only was performed on vivax samples

Reviewer #2: This manuscript from Han and colleagues describes the application of a microsatellite genotyping approach (HipSTR) to a large collection of Plasmodium falciparum and Plasmodium vivax malaria parasite short read whole genome sequencing datasets. Though microsats are common in these genomes, they have historically been paid less attention than SNPs due to difficulty in accurately calling microsat length polymorphisms from microsatellite loci. The authors demonstrate that HipSTR microsatellite calls are generally reliable and recapitulate or complement many selection and population structure patterns observed with SNPs.

This manuscript represents a solid body of work, and many in the malaria field will be appreciative to see a thoroughly-explored methodology for calling microsat variants in Plasmodium. However, the manuscript could be improved through attention to a few issues:

1) Line85: the authors mention a host of methods for calling microsat/STR variants, but only evaluate HipSTR in this manuscript. Can readers be confident that this is indeed the best tool for Plasmodium? Did the authors evaluate other approaches?

2) The Pf3K dataset includes a set of sequenced progeny from Plasmodium sexual crosses, which were very useful for identifying a gold standard set of variants for developing best practices for SNP calling with GATK. Did the authors consider using this approach for evaluating HipSTR calls, which could offer a more comprehensive validation than gel electrophoresis of a small panel of microsats as in Fig S1? Genome wide selection scans and analyses of population structure are not dependent on uniformly high accuracy of calls, but other applications could require highly accurate calls. Similar to how the Pf3K consortium delimited the ‘accessible’ regions of th P. falciparum genome for read alignment and accurate SNP calling, it could be very useful to the malaria field to produce a set of coordinates of microsats that would be expected to yield highly accurate calls based on their features and performance with cross progeny.

3) Line 270 and Figure 3: I don’t understand what is meant by the claim that STRs provide ‘greater resolution of distinct samples at the local scale’ than SNPs, considering that the analyzed samples do not come up local-scale geographic metadata below the level of country. The SNP vs. STR plots in Figure 3 look generally similar, and without metadata indicating the truth about precisely when and where samples were collected, how is it possible to claim that one UMAP plot or NJ tree provides more resolution than another?

4) The Discussion section is very long and spends much time simply recapitulating the Results. It would be helpful to better contextualize key results in this section. This manuscript generally shows that microsats/STRs can recapitulate many findings first observed with SNPs in P. falciparum or P. vivax. Can the authors further comment on why microsats should be profiled on a genome-wide scale? Some purported advantages of microsats over SNPs are not clear. For example, (line 559) why would microsats reveal recent signals of selection more quickly than SNPs? Their capacity for recurrent mutation would seem to make them less useful in general for tagging genomic variants under selection, and reason suggests that strong directional selection would result in sweep signals simultaneously in SNP and microsatellite markers.

Small points:

1) Figure 3 caption, part B is a trailing sentence.

Reviewer #3: see attached

**Have all data underlying the figures and results presented in the manuscript been provided?**

Reviewer #1: Yes

Reviewer #2: Yes

Reviewer #3: Yes

PLOS authors have the option to publish the peer review history of their article (what does this mean?). If published, this will include your full peer review and any attached files.

Reviewer #1: No

Reviewer #2: No

Reviewer #3: No

---

## [Decision Letter · Decision Letter 1]

23 Nov 2021

Dear Dr Bahlo,

Thank you very much for submitting your Research Article entitled 'Population-level genome-wide STR typing in Plasmodium species reveals higher resolution population structure and genetic diversity relative to SNP typing' to PLOS Genetics.

The manuscript was fully evaluated at the editorial level and by independent peer reviewers. The reviewers appreciated the attention to an important topic but identified some concerns that we ask you address in a revised manuscript

We therefore ask you to modify the manuscript according to the review recommendations. Your revisions should address the specific points made by each reviewer.

[LINK]

Yours sincerely,

Giorgio Sirugo

Associate Editor

PLOS Genetics

Hua Tang

Section Editor: Natural Variation

PLOS Genetics

Reviewer's Responses to Questions

**Comments to the Authors:**

Reviewer #1: The authors have carefully considered and responded to each of my original comments. The inclusion of genetic cross data is a great addition. My original concerns about the validation has been mostly addressed and the updates have added clarity. I particularly appreciate the lengthy addressing of comments regarding the circularity of the approach. I also appreciate the availability of code from the work making the analysis of new data feasible.

The remaining issue I have is with the novelty of the work. The work reanalyzes data from a large scale effort where microsatellites have also been called. Notably, the prior work focused on SNPs and indels, did not differentiate between other indels and microsats, and did not greatly delve into using indels to inform population structure or selection all areas focused upon here. The use of microsats to infer population structure of malaria parasites is very well established, they are widely used albeit often as much smaller panels, and localized patterns of selection (i.e. reduced expected heterozygosity) have been previously exploited to characterize positive selection. Conversely, there is great value is a robustly and specifically called set of microsatellite markers, and the authors do a wonderful job of demonstrating that they are powerful for understanding population structure and recent selection. The work is well placed in the context of other research, though a clearer presentation as to why we need a novel approach for calling microsats, and where the current data are an improvement over either genome-wide SNPs or smaller panels of microsatellites would address this.

Reviewer #2: The authors have provided a thorough response to my comments and those of the other two reviewers. The validation of the STR typing on the sexual cross data from the Pf3K project, in particular, instills much greater confidence in the accuracy of this approach.

The manuscript still fails to strongly connect on several fronts in terms of the utility of the findings. For example, Figure 3 still provides no evidence that STRs provide a higher resolution of population structure than SNPs, despite this claim comprising much of the title. Sup Figures 8 and 9 show this, but the use of NJ trees to evaluate populate structure in a sexually recombining eukaryote is not an apt or rigorous way to assess this.

Given the new capacity to perform sexual crosses in humanized mice, it could be interesting to relate this STR genotyping capacity to the capacity to fine-map QTL signals. So, in the discussion of the Pf3K cross progeny validation, it would be useful to note how many STRs were successfully genotyped and were segregating in each cross relative to SNP markers. Reporting the average MER % does not give a sense as to how much value STRs could add to this application.

As reviewer 1 notes, the Redmond et al. paper undertook a genome-wide analysis of STRs for the purpose of understanding clonal transmission dynamics. This HIPSTR-based approach could be very exciting for assessing the age of clonal lineages, or bifurcations in their transmission history, due to the expected faster rate at which de novo STR mutations should accumulate relative to SNPs. As a reviewer, I don't think it's fair to think to heap significant new analyses on a manuscript after it's already been revised once (and therefore I do not expect the authors to undertake an analysis of the Redmond data), but this is an example of an application for which this approach could uncontroversially outperform SNPs. Perhaps the authors could allude to this potential application.

Reviewer #3: See attached.

**Have all data underlying the figures and results presented in the manuscript been provided?**

Reviewer #1: Yes

Reviewer #2: Yes

Reviewer #3: Yes

PLOS authors have the option to publish the peer review history of their article (what does this mean?). If published, this will include your full peer review and any attached files.

Reviewer #1: No

Reviewer #2: No

Reviewer #3: No

---

## [Editor Report · Decision Letter 2]

14 Dec 2021

Dear Dr Bahlo,

We are pleased to inform you that your manuscript entitled "Population-level genome-wide STR discovery and validation for population structure and genetic diversity assessment of Plasmodium species" has been editorially accepted for publication in PLOS Genetics. Congratulations!

Yours sincerely,

Giorgio Sirugo

Associate Editor

PLOS Genetics

Hua Tang

Section Editor: Natural Variation

PLOS Genetics

Comments from the reviewers (if applicable):

**Data Deposition**

http://datadryad.org/submit?journalID=pgenetics&manu=PGENETICS-D-21-00692R2

**Press Queries**

---

## [Editor Report · Acceptance letter]

6 Jan 2022

PGENETICS-D-21-00692R2 

Population-level genome-wide STR discovery and validation for population structure and genetic diversity assessment of </i>Plasmodium</i> species 

Dear Dr Bahlo, 

We are pleased to inform you that your manuscript entitled "Population-level genome-wide STR discovery and validation for population structure and genetic diversity assessment of </i>Plasmodium</i> species" has been formally accepted for publication in PLOS Genetics! Your manuscript is now with our production department and you will be notified of the publication date in due course.

With kind regards,

Agnes Pap

PLOS Genetics

On behalf of:
